# "Focus on glaciers": a geo-photo exposition of vanishing beauty.

Giuliana Rossi, Gualtiero Böhm, Angela Saraò, Diego Cotterle, Lorenzo Facchin, Paolo Giurco, Renata Giulia Lucchi, Maria Elena Musco, Francesca Petrera, Stefano Picotti, Stefano Salon.

National Institute of Oceanography and Applied Geophysics -OGS, Sgonico, 34010, Italy

*Correspondence to*: Giuliana Rossi (grossi@inogs.it)

**Abstract.** Scientific research, respect for the environment, and passion for photography merged into an exceptional heritage of images collected by the researchers and technicians of the National Institute of Oceanography and Applied Geophysics - OGS. The images were taken during past scientific expeditions conducted all over the world to widen scientific knowledge in the fields of earth and ocean sciences, to raise environmental awareness and conservation of natural resources, and to mitigate natural risks.

In this paper, we describe a photographic exhibition organized using some of the OGS images to draw public attention to the striking effects of global warming. In the artistic images displayed, the glaciers were the protagonists. Their infinite greyish-blue shades and impossible shapes were worthy of a great sculptor, and the boundaries with rocks or with the sea were sometimes sharp and dramatic and sometimes so nuanced that they looked like water colours.

The beauty of the images attracted the attention of the public to unknown realities, allowing us to document the dramatic retreat of the Alpine glaciers and to show the majesty of the Arctic and Antarctic landscapes, which are fated to vanish under the present climate warming trend.

The choice of the exhibition location allowed us to reach a broad public of working-age adults, who are difficult to involve in outreach events. The authors of the images were present during the exhibition to respond to visitors' curiosity about research targets, the emotional and environmental context, and the technical details or aesthetic choices of the photographs.

## 1 Introduction

The route towards a sustainable world requires a profound change in the way we deal with the planet's resources, which will involve everyone: institutions, businesses, consumers and citizens will be called upon to collectively create a new model of development.

In September 2015, the United Nations General Assembly approved the Agenda 2030 for Sustainable Development, i.e., a plan of action that all countries (policy and citizen) have to respect in the coming years to achieve sustainable development by 2030 (United Nations, 2015). The Agenda 2030 is composed of 17 main Sustainable Developments Goals in areas of utmost importance for humanity and the planet. Action against climate change is at the core of Goal 13: *Take urgent action to combat climate change and its impacts*. In particular, target 13.3 suggests that countries "Improve education, awareness-raising and human and institutional capacity on climate change mitigation, adaptation, impact reduction, and early warning".

Limiting future global warming to 1.5°C requires rapid, far-reaching, and unprecedented changes in all aspects of society, but it would imply clear benefits to people and natural ecosystems while ensuring a more sustainable and equitable society (IPCC, 2013).

By the end of 2019, interest in climate change and the dangerous effects of present global warming had become very widespread. The actions of Greta Thunberg and the "Fridays for Future" movement played a primary role in increasing people's awareness and promoting public debates on this issue. In 2020, it became evident that an increasing number of people are making small but effective steps in the direction of plastic and emission reduction, energy saving, and environmental protection. The so-called 'Greta 'effect' led wealthy philanthropists and investors from the United States to donate almost half a million pounds to establish the Climate Emergency Fund (e.g., Taylor, 2019). The idea is to spread the money widely to many groups in relatively small increments for small but effective actions. However, just a couple of years ago, this topic was mostly ignored, notwithstanding the already high consensus among scientists about the anthropogenic impact on global warming (AGW), public opinion was not aware of it or denied its existence. The primary reason inducing people to deny AGW during public debates was the apparent lack of agreement between scientists (Cook et al., 2016, and references therein). We recognize that the problem of communication between scientists and the general public is an essential issue in many science fields. Lacchia et al. (2019) analysed the difficulties in communication between geoscientists and non-geoscientists. According to the study, public opinion about geosciences often focuses on the negative environmental impacts of geosciences activities (e.g., energy supply, mineral resource exploitation) rather than on their role in developing basic knowledge on our planet and for environmental protection. To overcome such prejudices and in agreement with the recommendations for science communication (Dahlstrom, 2014), Lacchia et al. (2019) recommended that other geoscientists also include their feelings, such as their motivations for the research, when outlining the impact of their own studies on knowledge and society to reach a broader audience. Effective communication with a large audience can ensure the broad support necessary for policy-makers to take the necessary actions once they are convinced of the firmness of the scientific results (Liverman, 2008).

The combination of science and art is becoming increasingly popular for improving the connection between science communicators and the public (e.g., Malina, 2010). Among the various strategies, photography is a practice of straightforward communication that is able to easily catch the interest of the public on critical questions. Furthermore, photography is the perfect combination of art and science because it naturally attracts people with different backgrounds or motivations. The proliferation of smartphones and software applications dedicated to image editing has made photography a common gesture in our lives. Every image can appear differently to the observers, eliciting emotional responses. Impressive photos can derive from either a scientific or artistic approach, but "*great photos often come from a combination of both art and science*" (Stone, 2017). The creation of a photograph requires emotion and imagination, although creativity and beauty can be engineered in post-production using editing software. Several photo exhibitions have been organized during the past a few years by professional photographers and artists worldwide in the framework of specific projects devoted to enlarging public awareness of the climate crisis by using the art to strengthen the message (e.g., Macromicro non profit Association,

2020) or to create an eye-opening performance to incite social and political change (e.g., Neudecker and Project Pressure Partnership, 2015). Other initiatives focused on integrating art and science, such as the Extreme Ice Survey (Balog et al., 2012; 2019), which produced a photographic book and a documentary movie that won an Emmy Award in 2014 (Chasing Ice, 2020). Online photographic collections from scientists are available through specific projects, such as those managed by the National Snow and Ice Data Center (2020), which is supported by NASA and the National Science Foundation (NSF) or by the USGS Northern Rocky Mountain Science Center (2019), which is focused on Glacier National Park. Another photographic repository collected by scientists is managed by the European Geoscience Union (EGU, 2019). Every year, during the EGU General Assembly, after a contest to choose the most beautiful photo, the photos with the most votes are granted, printed and freely distributed as postcards to reach a wider public and show the beauty of our planet.

Drawing on the rich image collection of the OGS, which is scientifically engaged all over the planet, we decided to organize a photographic exhibition focused on the glaciers and ice sheets distributed at different latitudes to convey a strong message to the public on the devastating effects of climate change on our planet, which aligns with the recommendation of the Agenda 2030, and in particular with the already mentioned specific target 13.3 of the Sustainable Development Goal n. 13, "*Climate Action*".

Ice sheets in polar areas and mountain glaciers above 2500 m have been shown to react particularly rapidly to the present climate warming (Shepherd et al., 2018; 2020). For this reason, we specifically focused on images of glaciers, ice caps, and icebergs as an efficient way to communicate the perception of the fragility of such environments, which are presently jeopardised by climate changes. The originality of our exposition, compared with the ones mentioned above, is that the authors are scientists involved in scientific activities during research cruises and not professional photographers. Our goal, in fact, was to close the gap between research and society: the exhibition became a way to bring scientists near the public, and specifically, working-age adults, in an environment usually unrelated to science. The images were taken during the scientific research activities on research vessels or in the field, and they reflect the intimate attitude and the sense of wonder of human beings in front of the supreme beauty of nature, combined with the artistic side of the scientist. During the exhibition, the visitors were able to satisfy their curiosity on the research aspects, the context in which the pictures were collected, the technical photographic details, and specific aesthetic choices. This paper presents a summary of this experience, which impacted both the authors and the visitors.

## 2 OGS mission and strategic view

The National Institute of Oceanography and Applied Geophysics - OGS is a public research institute supported by the Italian Minister of University and Research (MUR). The institute is active in the research fields of geosciences of the solid earth and oceans to increase scientific knowledge, mitigate geohazards, exploit and conserve natural resources, and raise environmental awareness from a sustainable development perspective. The OGS employs a staff of approximately 300

people, and it promotes research through the joint use of its main research infrastructure (i.e., research vessels and aircrafts and onshore and offshore monitoring networks).

Due to its long-term collaboration with the energy industry, the OGS developed high-technology competence and skills in acquiring, processing, interpreting, and modelling onshore (surface and borehole) and offshore geophysical and oceanographic data. The interdisciplinary character of the OGS has allowed it to produce fundamental contributions to the challenges of the present time. In particular, OGS research activities have enabled assessment of the past and current state of the environment to define future scenarios, considering natural forcing and human activities, and to exploit the most advanced computing technologies for climate model data production and analysis at the local or global scale. Further, multidisciplinary studies contributed to the definition of strategies to reduce the greenhouse effects of $CO_2$ through its sequestration in geological storage.

In agreement with the general principles of the European Charter for Researchers and Code of Conduct, the OGS is extensively engaged in dissemination and communication activities. The OGS communication strategy includes the organization and participation in public events to maintain an open dialogue with stakeholders, citizens, and young people and to share knowledge and outcomes in support of society. Among these, several dissemination events were performed within international initiatives, such as the European Researchers' Night, the Pint of Science Festival, or local initiatives, such as the Trieste festival of the scientific dissemination (NEXT), or appointments with science in the historical Cafès of Trieste.

## 3 Visual communication and the exhibition

The main elements of the communication process derive from the models of Shannon (1948) and Berlo (1960). The main elements are the *sender* (the person transmitting the message), the *receiver* (the person receiving the message), the *message* (the communication subject), the *channel* (the communication vehicle), and the *context* (where, how, and when the message is sent). The general difficulties of the scientific community in transferring their research results and insights are well known, and this is particularly true when the *message* concerns environmental issues and communication is addressed to the general public or political class. Photographic books and exhibitions provide precious support to convey knowledge because they allow the images to be observed and pondered more slowly. Photography, which is a *channel* of communication, uses a universal language that can reach a large number of people, especially today, when the bulk of the information is mainly conveyed through images. Indeed, photography is much more immediate than text, and it provides a quantity of information that can be perceived at one glance and be quickly memorized. Therefore, we identified photography as a powerful and efficient *channel* for communicating the need to protect specific environments that are strongly endangered by global change. During the selection of the photos for the exhibition, preference was given to high-quality images evoking emotions on natural beauty that could be lost.

Among the elements of visual communication, the *context* is as important as the *message* and the *channel*. In our case, the photographic exhibition was set up in a popular, often crowded workplace to reach the widest range of visitors, including working-age people who generally do not attend public conferences or other dissemination events. The exhibition itself, intended as an ensemble of multiple images of easy and quick reading, strengthened the *message*, even through a short, often rushing view.

## 4 The exhibition

The photographic exhibition "Focus on glaciers" took place in Trieste during October 2016 in the lobby of the early-XIX century neoclassical palace, initially the seat of the stock exchange established by Maria Theresa of Habsburg and now headquarters of the Trieste Chamber of Commerce. The venue was specifically chosen to attract people who cross the lobby daily for work activities. The exhibition was scheduled among the public events foreseen for the *Settimana del Pianeta Terra* (*The Week of the Earth Planet,* Figure 1), a scientific festival spread throughout the Italian territory aiming to promote the geosciences and to increase public awareness for the reduction of natural risks. The photographs for the exhibition were selected after an OGS internal call to collect images focused on glaciers, acquired during scientific expeditions and field trips in the polar areas, or other relevant regions. Indeed, the OGS researchers and technicians throughout the years collected an exceptional heritage of high-quality images. For each photograph, the authors had to provide the information about the place, the year and season, the scientific context, and a comment on the motivation, emotional context, and technical details. A committee, formed by geoscientists who were experts in photography and communication skills, selected the images that were most suitable for the exhibition following the principles expressed in Section 3. Aesthetic and technical criteria mainly guided the choice of the photographs, but particular attention was also paid to the message that the image could convey to the *receiver*. The committee received 130 photographs, from which 26 images were chosen for the exhibition, corresponding to approximately 20% of the original photographic set. The photographs mainly illustrated the two polar regions, as well as the Alps and other mountainous regions. The exhibition was freely accessible to the Chamber of Commerce's visitors and employers and therefore to working-age adults (18-64 years). At the exhibition opening, as well as on the occasion of other conferences related to the Earth Planet public event, the authors of the photographs were present and directly interacted with the public (Figure 2).

In the following, we present the areas where the photographs were taken, grouped into two main domains: the polar regions and mountain chains (Figure 3).

### 4.1 The polar regions

Polar amplification (i.e., the exacerbated effects of climate change at the poles with respect to the rest of the hemisphere) has been well documented within climate change studies through both historical and instrumental observations and model simulations. The causes of this effect are still a matter of discussion (see Stuecker et al., 2019, and reference therein). In

Antarctica, the total average ice loss per year was 43 gigatons during the 1992–2002 decade, but it sharply accelerated to an average of 220 gigatons per year from 2012–2017 (Shepherd et al. 2018). The Arctic region is warming even more rapidly: the Svalbard Archipelago, which is located between 74° and 81° north latitude, has experienced the fastest air temperature increases in the last three decades (Nordli et al., 2014), and climate model projections showed that this trend would continue until the end of the XXI century (Førland et al., 2011). Consequently, the accelerated mass loss of the glaciers in western Svalbard implied an increased contribution to sea level (Kohler et al., 2007; Nuth et al., 2010). In a few years, the Arctic sea ice will disappear during the summer season, opening new commercial and tourist routes through the North Pole: the routes from the Far East to Europe can be shortened by sailing along the Siberian coast instead of via the Suez Canal. Furthermore, easy access to the Arctic Ocean will make the large oil fields of this area very attractive, with the additional potential environmental risk represented by their exploitation. On the other hand, the exceptional melting and retreat of the ice shelf in Ross Bay in Antarctica, documented by OGS researchers in 2018, enabled the acquisition of important information in unexplored areas that were inaccessible during the past years. The white ice coverage in polar areas, either as sea ice or continental ice sheets, helps to regulate the Earth's climate by reflecting most solar energy back to space, whereas the dark oceans/seas absorb most of the solar radiation, further contributing to Earth and climate warming.

Earth's climate warming affects not only ice extension and glaciers but also human lifestyles. In particular, Nordic peoples, such as Eskimos, risk seeing their livelihoods strongly compromised, and animal species such as polar bears are threatened with extinction (Giovannini and Speroni, 2019). The Svalbard Global Seed Vault, which hosts and protects world seed varieties to prevent accidental loss of diversity, is now in potential danger.

### 4.1.1 Antarctica

The OGS has continuously developed research in Antarctica since 1988 with funding from the *Programma Nazionale di Ricerche in Antartide* (PNRA), through the MUR and from Europe within the programmes of the *Scientific Committee for Antarctic Research* (SCAR). OGS researchers and technicians have developed considerable skills in the geological, geophysical, and biological fields during many geophysical/oceanographic/geological research campaigns in Antarctica with the research vessel (RV) *OGS-Explora*, the RV Italica, and other research vessels belonging to OGS's international partners. During 2019, OGS acquired the RV "*Laura Bassi*", an icebreaker class ICE 05 E0 that is managed in cooperation with the *Consiglio Nazionale delle Ricerche* (CNR) and the *Agenzia Nazionale per le nuove tecnologie, l'energia e lo sviluppo economico sostenibile* (ENEA). Furthermore, the OGS participated in several onshore international projects in remote field operations at the Italian Bases Mario Zucchelli and Concordia; in collaboration with the Argentine Antarctic Institute, it has managed the Antarctic Seismographic Argentinian-Italian Network since 1992 (Russi et al., 2010).

During Antarctic expeditions on research vessels, researchers, technicians, and crew stay on board for approximately two months, sharing every moment of life during data acquisition and convivial breaks. They bring home the feeling of a magical experience despite the often harsh environment and the hard work, together with many photographs of beautiful landscapes

crossed during the cruise or the fieldwork. Our exhibition included images from the XXI, XXVIII, XXIX, XXX, XXXI campaigns to Antarctica (Figure 3a, Figures 4-7). The icebergs, seracs, and cliffs of ice fronts were the main photographic subjects (Figures 4-6), with the alternation of white snow and ice and blue ice generated by the compression of air bubbles incorporated in the ice (Figures 4a, d; 5; 6a-c). Figure 7 shows the sole animated subject of the whole exhibition: a lonely, small penguin drifted on an iceberg in the middle of Antarctica.

### 4.1.2 Svalbard islands

The OGS started its research activity in the Svalbard archipelago in 1971 with an exploration-seismic cruise funded by Norsk-Agip (Deluchi, 2013). Since 2001, OGS researchers have been involved in several research cruises (four with the RV OGS-Explora but also with Norwegian, German, and Spanish vessels, also thanks to the Eurofleets EC-FP7 project), as well as on land, within international projects (Figure 3 b), often below the umbrella of the *International Arctic Science Committee* (IASC). The Svalbard treaty bans military activities in the Arctic but not research bound to mining or hydrocarbon exploration. This was the case of the *Paleokarst* research project (*Paleokarst Reservoirs: An integrated 3D approach to heterogeneity, reservoir and seismic modelling)* jointly funded by industrial partners and the Norwegian Research Council, which aimed to study with geophysical methods the structure and physical properties of an onshore analogous to the reservoirs at depths below the Barents seafloor. Within this project, the researchers conducted research while living on a remote camp onshore in sight of the mouths of several glaciers (Figures 8b, d) and had to apply strategies to prevent polar bear attacks. This project was followed by the PNRA project –"*Integrated Methods to study PERmafrost characteristics and Variations In an Arctic natural laboratory (Svalbard)- IMPERVIA*", which was another field work campaign focused on the study of permafrost (Rossi et al., 2018). Other projects developed offshore from the western and southern margin of Svalbard have focused on the present and past oceanographic characteristics of the Western Spitsbergen Current (the northern branch of the warm North Atlantic oceanic current) and its impact on the dynamics of the paleo-Svalbard-Barents-Sea ice sheet finalized to palaeoclimatic reconstructions. Further research activities targeted the identification of biological oases associated with seepage activities in relation to the presence of gas hydrates developing at the subseafloor. In the last two cases, the photos were collected from research vessels during the transfer to different study areas or sailing back to land, after several days or months of on-board activity, often under harsh climatic conditions, rough sea or completely blind in the thick fog or in the winter darkness, with the snow-covered land appearing like a mirage (Figures 8a, c; 9a, b).
In a field camp in Skanskbukta Bay (Figure 3b, Figure 9c), with the basecamp encircled by breath-taking mountains, small waterfalls and creeks, the OGS researchers witnessed several huts as vivid memories of human activities at the beginning of the last century.

### 4.2 Mountain chains: the Alps and the Rocky Mountains

During 2019, for the first time, the International Panel on Climate Change (IPCC) released a report on the present impacts of climate change on the world's mountain environments. The surface air temperature in the mountains of western North

America, the European Alps, and High Mountain Asia increased at an average rate of 0.3°C per decade during the last three decades, therefore outpacing the global warming rate (IPCC, 2019). The snow-coverage duration, thickness, and extent decreased by an average of 5 days per decade, especially for those at lower elevations. From 2006–2015, the mass change of the glaciers in most of the mountain regions, excluding the polar areas (Canadian and Russian Arctic, Svalbard, Greenland, and Antarctica), was approximately -490 ± 100 kg $m^{-2}yr^{-1}$ (123±24 Gt $yr^{-1}$). The regionally averaged mass budgets were mostly negative (less than -850 kg $m^{-2}yr^{-1}$) in the southern Andes, Caucasus and Central Europe and least negative in High Mountain Asia (-150±110 kg $m^{-2}yr^{-1}$). Sparse and unevenly distributed measurements have shown a progressive increase in the permafrost temperature, with a shift of 0.19±0.05°C on average for approximately 28 locations in the European Alps, Scandinavia, Canada, and Asia during the past decade.

### 4.2.1 The Alps

Between the end of the 19[th] and the beginning of the 21[st] century, the average air temperature on the Alps rose by approximately 2°C, i.e., more than twice the temperature increase observed throughout the Northern Hemisphere. Over the same period, the rainfall mass recorded an increasing trend in the northern part of the Alps and a decreasing trend in the southern sector.

Since the end of the Little Ice Age (LIA, ca. 1850 in Europe), a general retreat of the glaciers in the Alps occurred, although it was locally interrupted by two short-lived phases of re-advance, which occurred during the 1920s and 1970s. However, it has been estimated that the glacial area in the Alps has been severely reduced by approximately half since the end of LIA and that the rate of reduction has considerably accelerated since the 1980s, especially on the southern side of the chain.

According to the last cadastre of the Italian glaciers (Smiraglia and Diolaiuti, 2015), over as few as fifty years, the total area has decreased from 527 to 368 square kilometres, leading to the extinction of 180 glaciers. Nigrelli et al. (2015) related the recent glacial shrinking with the climatic variations documented by the meteorological stations, providing an accurate picture of the rapid regression of the glaciers, and quantifying the relationships between climate and glaciers.

However, we can hypothesize that, at least in some cases, the combined action of the temperature increase and precipitation decrease that occurred after 1980 influenced the evolution of glaciers. According to the present rate of glacier decline and according to the observed climatic warming trends, the glaciers in the Italian Alps are expected to disappear by 2050 (Santin et al., 2019).

In the frame of the PNRA project "*Subglacial lake exploration in the Whillans Ice Stream region (West Antarctica) - WISSLAKE*", the OGS researchers performed geophysical tests on the Alpine glaciers to evaluate the feasibility of the applied methods in quantifying the glacier thickness and structure (Figures 10a, b, c; Picotti et al., 2017). The geophysical methods have been used on the glaciers of the Adamello and Ortles-Cevedale massifs (Italy) and the Bernese Oberland Alps (Switzerland), as well as on the Whillans Ice Stream (West Antarctica). Many sites were inspected along the Alpine chain to find suitable sites for the application of such techniques. The retreating glaciers bared their surface structure and crevasses, creating fascinating graphic effects (examples from Mont Blanc, Figures 11 b, c).

### 4.2.2 Canada

The annual and seasonal average temperatures across Canada increased during recent decades, with the most significant warming occurring during the winter seasons. In particular, from 1948-2016, northern Canada recorded an increase of 2.3°C compared to the 1.7°C of the whole country.

Unlike the Alps, in Canada, the precipitation averaged over the country has increased by approximately 20% from 1948 to 2012 (Vincent et al., 2015). Nevertheless, during 2007, the glaciers' volume loss was estimated as much as $22.48 \pm 5.53$ km$^3$ yr$^{-1}$, but such a high rate has recently further accelerated. A glacier such as the Peyto Glacier in the Rocky Mountains and part of Banff National Park has lost approximately 70% of its mass in the past 50 years. OGS researchers also performed site inspections in Banff National Park to further test the geophysical methods applied to the study and monitoring of glacier retreat around the world (Figure 11 a).

### 5 Discussion and Conclusion

The OGS exhibition "*Focus on glaciers*" used the beauty of the images and the impression of majesty and peace that the glaciers can inspire to visitors to transmit the message of environmental fragility and its protection. In recent years, the OGS has already participated in photographic exhibitions of research activities (*Night of Researchers* in Trieste, in 2013; *30 years of the Italian research programme in Antarctica* in Rome in 2015), but the *Focus on glaciers* exhibition was the first attempt by the OGS to use research photography to animate people on climate change themes. The authors of the photographs are research scientists involved in offshore and inland scientific activities as well as effective actors in artistic production, following the means by which art and science can work together (Malina, 2010), even if they are not professional photographers. The message that some prompt actions can still help to reduce the climate crisis was conveyed through the emotion that streamed from the pondered view of single, high-quality images representing the vanishing beauty of glaciers. As our exhibition was an a posteriori collection of photographs taken during short-term scientific offshore expeditions or inland campaigns, almost never in the same place, it was not possible to document the temporal transformation of the studied areas as a consequence of climate warming. However, we judged worthy of using the large number of photographs witnessing the magnificence and grandeur of a fragile landscape that is in danger of extinction. The criterion of high technical quality and strong emotional impact, drove the accurate selection of the images. This choice was aimed at obtaining a fast and immediate reading of the message by the *receiver*. This was the case for the collapsed icebergs shown in Figures 5a, 5d, and 6a; the blue ice floating in the rough sea (Figure 4d); or the lonely penguin set on a drifting iceberg (Figure 7) as an emblematic symbol of the living species in danger of extinction due to the climate crisis. Figure 8d and the graphical effects transmitted by Figures 11b and 11c dramatically document glacier melting and the possible desolation of the future landscape. The contemplation of the 26 photographs as a whole produced a strengthening of the *message* that the viewer perceived even in a fleeting passage through a public, crowded place. Therefore, the exhibition became the way to bring scientists closer to the public, taking specifically into consideration the working-age adults (18-64 years) in an

environment typically unrelated to science. The location of the lobby of the Chamber of Commerce of Trieste appeared to be an excellent choice: approximately 100 people every day crossed the location for their business, so we could easily quantify the engaged audience of approximately 2000 people from different social classes, cultural levels, and nationalities. Moreover, during the opening of the exhibition and on the occasion of some other conferences related to the Earth Planet public event, approximately 250 people had the unique opportunity to interact directly with the authors of the photographs. The most common question asked addressed the modality of the ongoing climate changes and the immediate impact on the present lifestyle. The authors had the duty to calibrate their answers in order to convey a simple but strong message without the use of complicated scientific or technical language or, worse, without making people feel powerless on the action on or mitigation of the climate crisis. In contrast, the very important message to convey was the necessity of acting fast and the possibility of success. Vivid conversations occurred near the panels hosting the artwork, while the visitors were delighted to satisfy their curiosity on either the technical aspects of the research, the development of the climate change studies, and/or the context during which the geoscientists took the photographs, or about more technical photographic details such as the camera exposure, the possible post-processing work, or aesthetic choices. Surprisingly, some technical questions regarded not only the topic of climate change but also the geology of polar areas and the geomorphology of glaciers, adding further scientific value to the artistic quality of the images.

The feedback received on the exposition accomplished Dahlstrom's recommendations (2014) and confirmed the observations made by Lacchia et al. (2019) about the importance of including emotional or challenging aspects during science communications such as research motivation or descriptions related to logistics, routine duties and lifestyle in extreme contexts. We believe that the choice of showing images of environments closer to our heritage, such as the Alps, facilitated the researchers in the delicate and difficult attempt of transmitting the message that the climate crisis is a real problem affecting all of us and that each small contribution from everybody can make a difference. The slow but inexorable vanishing of glaciers is striking evidence that global warming is effectively occurring here and now, and it will probably deeply affect the way our entire society will act in the future. Global warming is an entity of such vast temporal and spatial dimensions that is so interconnected with human activities that it seems to defy not only our control but also our understanding. Our concepts of the world and the environment must necessarily change to allow new awareness and to promote a sustainable and respectful coexistence between human society and nature. Communication activities, such as the *Focus on glaciers* exhibition, and other outreach actions promoted by OGS and other institutions, are critical to highlight the problem and make it relevant to the general public. The debate about climate change communication strategies is still active, and catastrophic frames are controversial (König and Jucks, 2019).

The exhibition project is still active: the pictures are presently displayed on OGS premises, and our colleagues are strongly encouraged to collect new images during their scientific expeditions to upgrade the exhibition. This experience may be further stimulated within the research community to keep track of and record the rapid changes occurring in the Earth's glaciers. The exhibition "Focus on glaciers" can be considered the first event for the OGS of a new way of communicating on the themes of climate change or on other themes of utmost importance for our society. Researchers can develop

alternative topics on the basis of the pictures collected during routine work that can be exposed through future similar exhibitions. Moreover, adding multimedia support, also showing moments of life during fieldwork or episodes related to the scientific campaigns, would be of importance to catch the visitor's attention and communicate more effectively. In the course of future events, we will further involve visitors through short surveys to verify whether the transmitted message was easily accessible and the level of awareness obtained after the visit of the exhibition.

**Author contributions.**

GR, GB and AS conceived the idea of the exhibition and wrote the paper. RGL, SP and SS read the paper and provided comments and corrections for improvements. DC and GR realised the maps of Fig. 3. GR and AS composed all the other Figures of the paper. GR, GB, DC, LF, RGL, MEM, SP, SS provided the images selected for the exhibition. PG and FP cured the installation of the exhibition and the advertising of the event.

**Special issue statement**

This article is part of the special issue "Five years of Earth sciences and art at the EGU (2015– 2019)". It is a result of the EGU GeneralAssembly 2016, Vienna, Austria, 17–22 April 2016.

**Competing interests**

The authors declare that they have no conflict of interest.

**Acknowledgements**

We warmly thank all the colleagues who sent us the photos from their expeditions to glaciers and polar areas. We are grateful to the Camera di Commercio, Industria, Artigianato e Agricoltura Venezia Giulia for hosting the exhibition in its premises.
We are indebted to Mariele Neudecker, an anonymous referee, and the editor Francesco Mugnai for their constructive and stimulating comments, which led to significant improvement of this paper.

**Financial support**

This research has been supported by Regione Autonoma Friuli Venezia Giulia (project "Diverso - Divulgazione e ricerca per un futuro sostenibile")

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

**Figure captions**

Figure 1: A sample of the flyer that reported some of the events organized by the OGS during the Settimana del Pianeta Terra (Planet Earth Week, https://www.settimanaterra.org). The opening of our exhibition "Obiettivo Ghiacciai: una bellezza che sta scomparendo" took place on October 17[th], 2016.

Figure 2: Photos taken during the exhibition.

Figure 3: Maps of the geographical domains where the pictures of the exhibition were taken. a) Antarctica; b) Spitzbergen island in the Svalbard Archipelago; c) the Alpine chain; d) the Rocky Mountain chain in Canada (for the topography Bright Earth eAtlas base map v1.0 (AIMS, GBRMPA, JCU, DSITIA, GA, UCSD, NASA, OSM, ESRI), ©  AU 3.0.).

Figure 4: Icebergs in Antarctica. a) Iceberg, XXI PNRA Antarctic expedition, project "*Western Ice Sheet Evolution –WISE*"; b, c) Sea ice view during shipping (Ross Sea). XXI PNRA Antarctic expedition, project WISE; d) Floating blue iceberg (Ross Sea). XXVIII PNRA Expedition, project "*Paleomagnetism of sedimentary cores from the Ross Sea outer shelf and continental slope-ROSSLOPE II*".

Figure 5: Icebergs and ice tongues in Antarctica. a) Collapsed iceberg (Ross Sea). XXIX PNRA Expedition, ROSSLOPE II project; b) Iceberg wall (Ross Sea). XXI PNRA Antarctic expedition, project WISE; c) Floating blue iceberg (Ross Sea). XXVIII PNRA Expedition, ROSSLOPE II project; d) Drygalski ice tongue (Ross Sea). XXXI PNRA Expedition, project "*Holocene climatic fluctuations in sub-millennial recorded in sedimentary sequences expanded the Ross Sea –HOLOFERNE*".

Figure 6: Antarctic landscapes. a-c): XXVIII PNRA Expedition, ROSSLOPE II project. a) Iceberg stacked in Cape Hallett (Ross Sea); b) Campbell glacier detail (southwestern Ross Sea); c) Floating blue iceberg (Ross Sea); d) Drygalski ice tongue (Ross Sea). XXXI PNRA Expedition, HOLOFERNE project.

Figure 7: A lonely penguin on a drifting iceberg (Ross Sea). XXI PNRA Antarctic expedition, project WISE.

Figure 8: Svalbard landscapes (Svalbard archipelago, Norway). a) Longyearbyen Bay, Tundra landform. R/V Polarstern expedition PS99-1a, Eurofleets2 project "*Bottom currents in a stagnant environment- BURSTER*"; b) A view from the Wordiekammen plateau towards the Ebbabreen, with the nunatak Bastonfjellet, Paleokarst project; c) Front of the Bellsund ice stream (SW Svalbard). RV Ian Mayen 2009 expedition, University of Tromsø-UiT, "*Glaciations in the Barents Sea Area –GLACIBAR*" project; d) From the Wordiekammen plateau towards the Petunia Bukta, with the waters from Horbyedalen and Ebbadalen, Paleokarst project.

Figure 9: Svalbard landscapes. a) Hornsund Fjord, Spitsbergen. RV G.O. Sars, expedition 191, "*Present and past flow regime on contourite drifts west of* Spitsbergen *Area- PREPARED*" Eurofleets2 project; b) Ice coverage of the Svalbard 'Islands' northwestern coast. R/V OGS Explora, "*Petroleum Assessment of the Arctic North Atlantic and adjacent marine areas- PANORAMA*" project; c) Skanskbukta Bay (on the left), Billefjorden (centre) with Bünsow Land cliffs (front). Field trip "*Poli Arctici Skanskbukta basecamp*", the "*Northern Rangers*" group.

Figure 10: Mountain landscapes. a) Piz Bernina (Italy); b, c) PNRA-WISSLAKE project: b) Monte Rosa (Italy); c) Giant Glacier, Mont Blanc (Italy).

Figure 11: a) A glacier on Mount Assiniboine, British Columbia, Canada. Field trip in the framework of the SEG 2009 Summer Research Workshop on "*CO$_2$ Sequestration Geophysics*"; b, c) A minor glacier in the Mont Blanc group (Italy). Field trip in the frame of the Near Surface Geoscience 2015 - 21[st] European Meeting of Environmental and Engineering Geophysics.

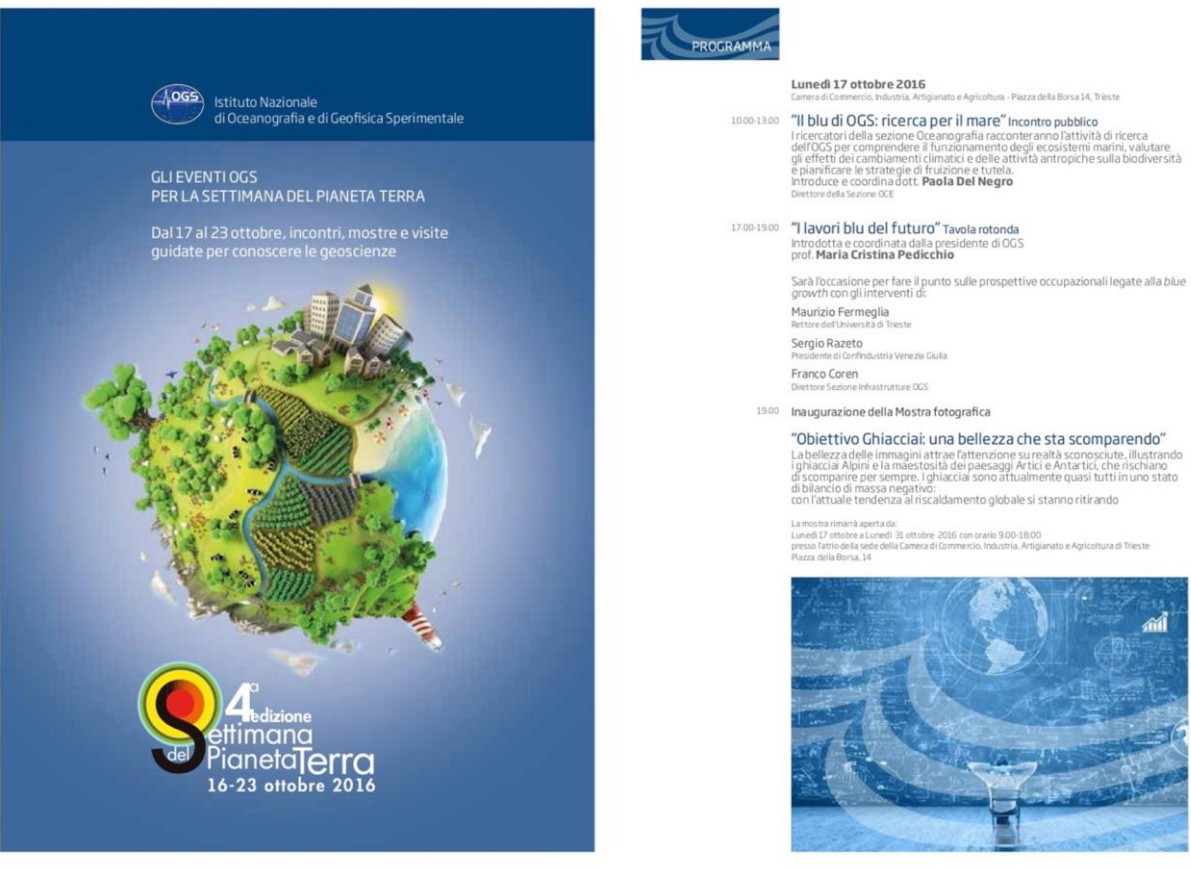

Figure 1: A sample of the flyer that reported some of the events organized by the OGS during the Settimana del Pianeta Terra (Planet Earth Week, https://www.settimanaterra.org). The opening of our exhibition "Obiettivo Ghiacciai: una bellezza che sta scomparendo" took place on October 17[th], 2016.

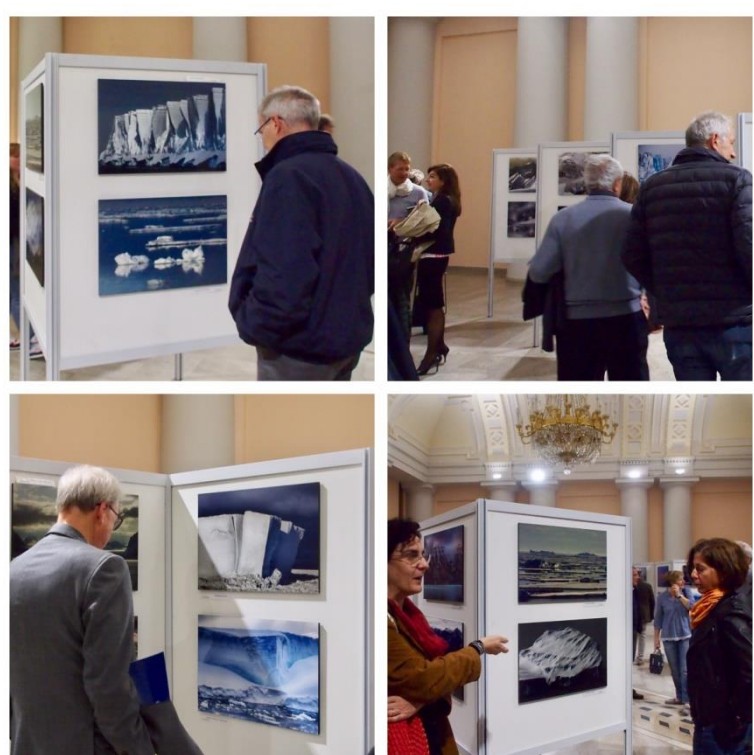

Figure 2: Photos taken during the exhibition.

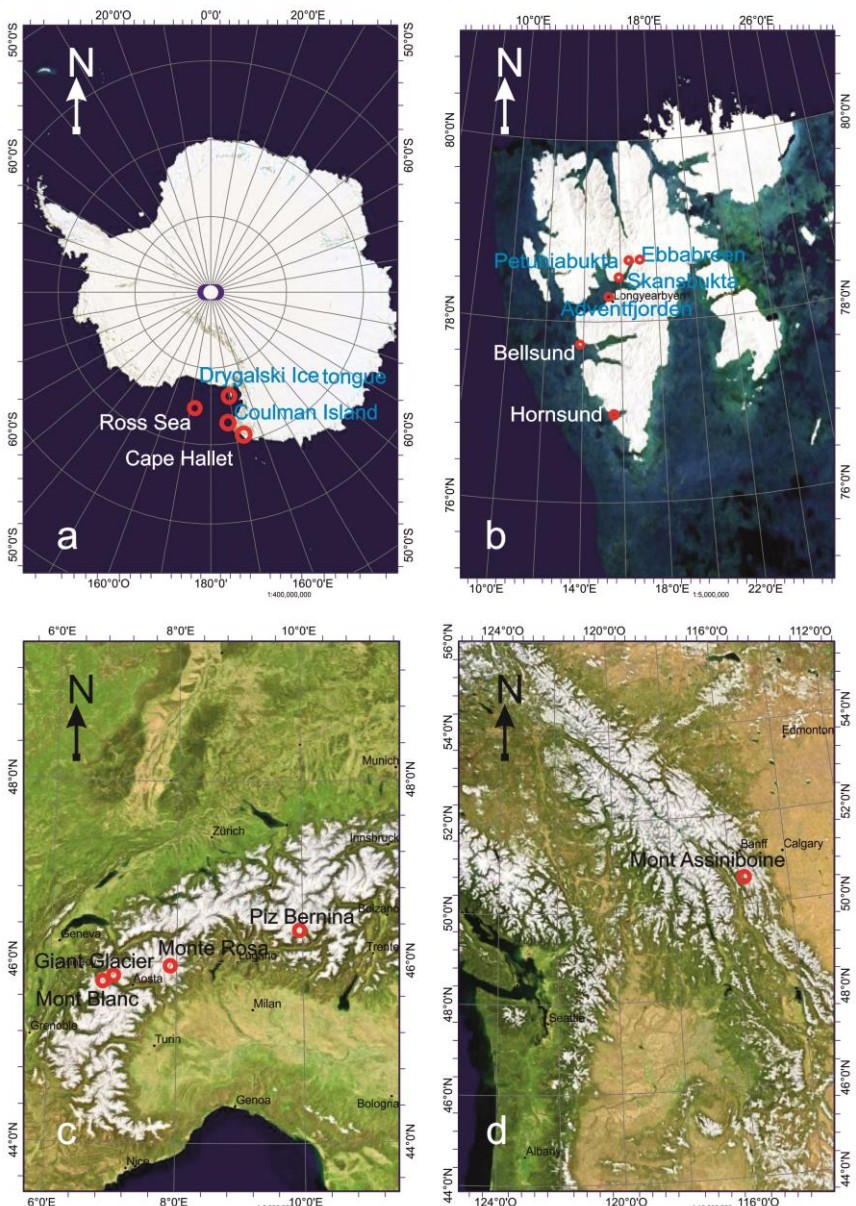

Figure 3: Maps of the geographical domains where the pictures of the exhibition were taken. a) Antarctica; b) Spitzbergen island in the Svalbard Archipelago; c) the Alpine chain; d) the Rocky Mountain chain in Canada (for the topography Bright Earth eAtlas base map v1.0 (AIMS, GBRMPA, JCU, DSITIA, GA, UCSD, NASA, OSM, ESRI), CC BY AU 3.0.).

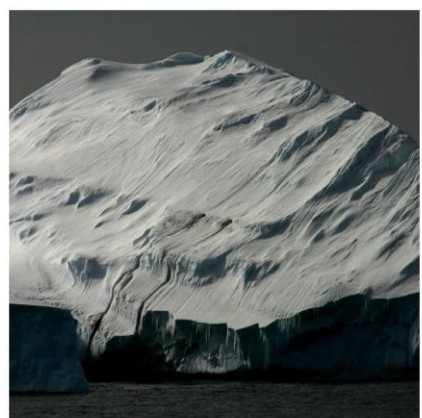 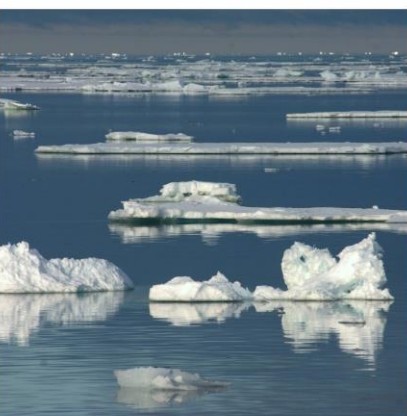 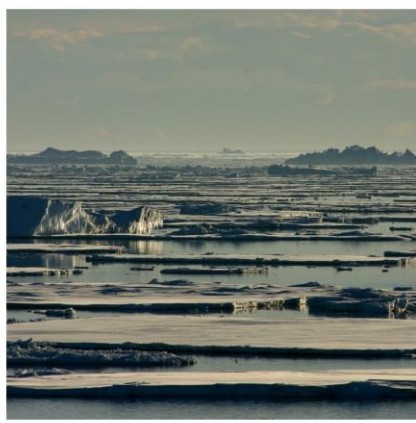

Figure 4: Icebergs in Antarctica. a) Iceberg, XXI PNRA Antarctic expedition, project "*Western Ice Sheet Evolution –WISE*"; b, c) Sea ice view during shipping (Ross Sea). XXI PNRA Antarctic expedition, project WISE; d) Floating blue iceberg (Ross Sea). XXVIII PNRA Expedition, project "*Paleomagnetism of sedimentary cores from the Ross Sea outer shelf and continental slope-ROSSLOPE II*".

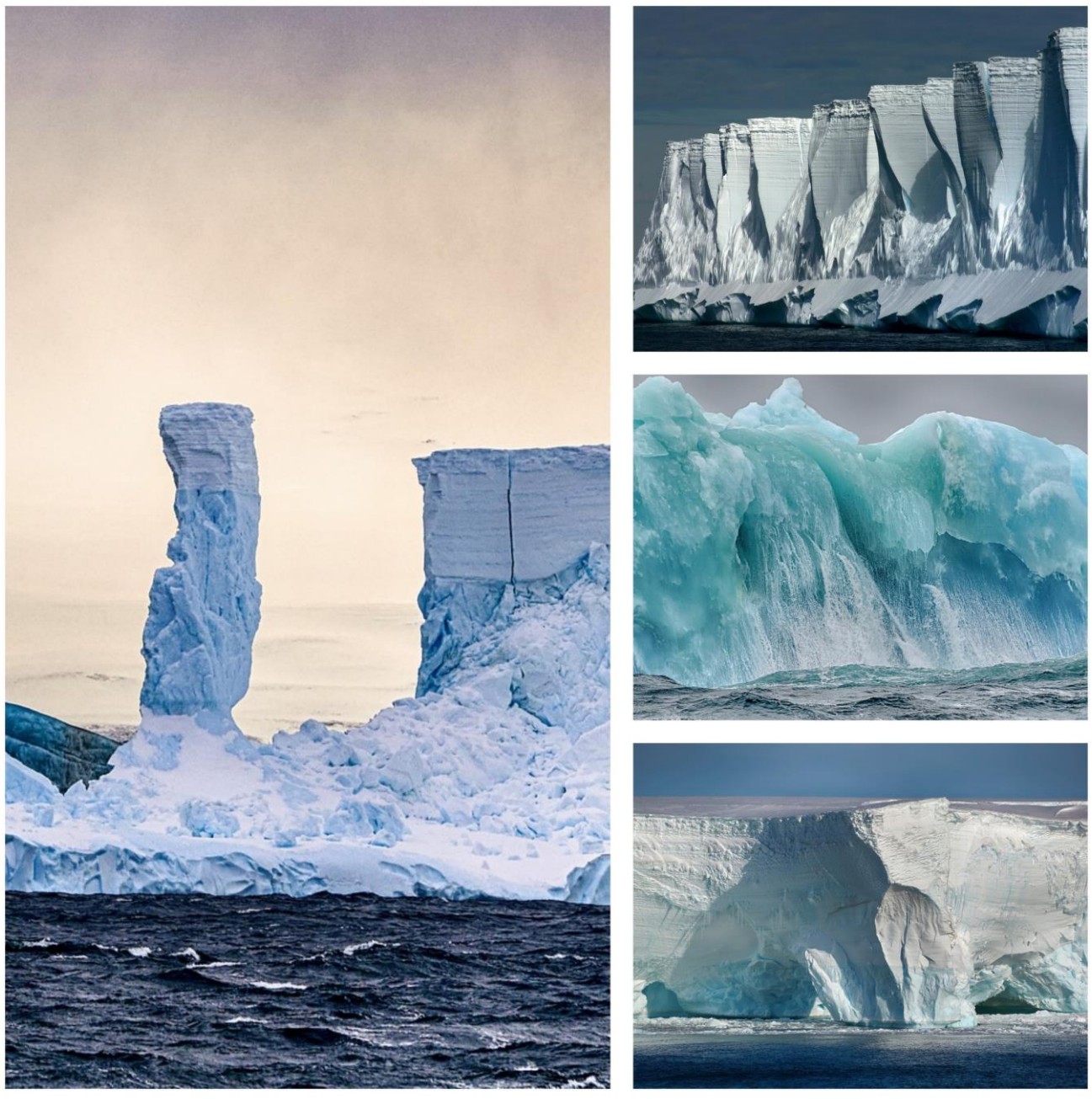

Figure 5: Icebergs and ice tongues in Antarctica. a) Collapsed iceberg (Ross Sea). XXIX PNRA Expedition, ROSSLOPE II project; b) Iceberg wall (Ross Sea). XXI PNRA Antarctic expedition, project WISE; c) Floating blue iceberg (Ross Sea). XXVIII PNRA Expedition, ROSSLOPE II project; d) Drygalski ice tongue (Ross Sea). XXXI PNRA Expedition, project "*Holocene climatic fluctuations in sub-millennial recorded in sedimentary sequences expanded the Ross Sea –HOLOFERNE*".

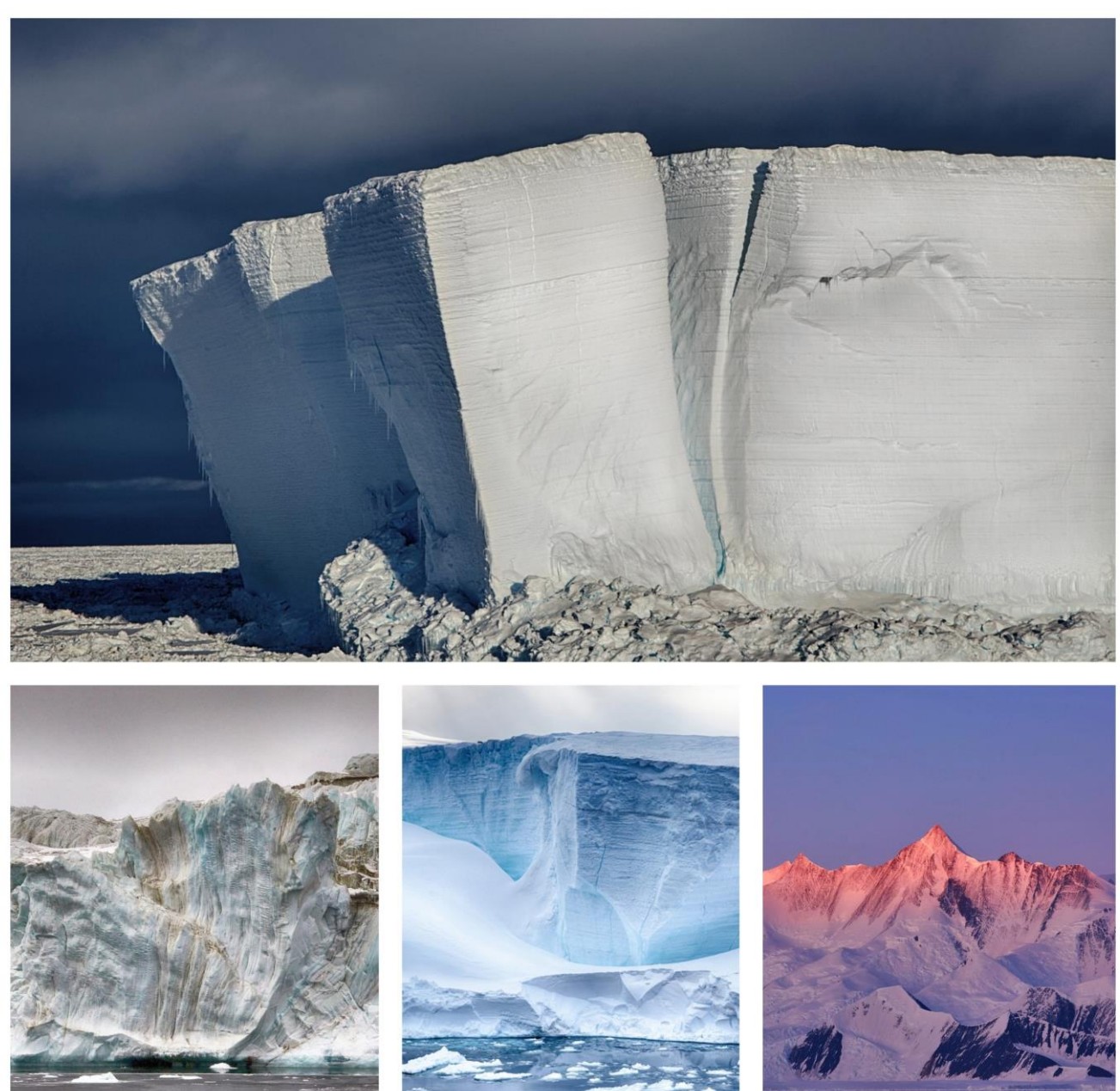

Figure 6: Antarctic landscapes. a-c): XXVIII PNRA Expedition, ROSSLOPE II project. a) Iceberg stacked in Cape Hallett (Ross Sea); b) Campbell glacier detail (southwestern Ross Sea); c) Floating blue iceberg (Ross Sea); d) Drygalski ice tongue (Ross Sea). XXXI PNRA Expedition, HOLOFERNE project.

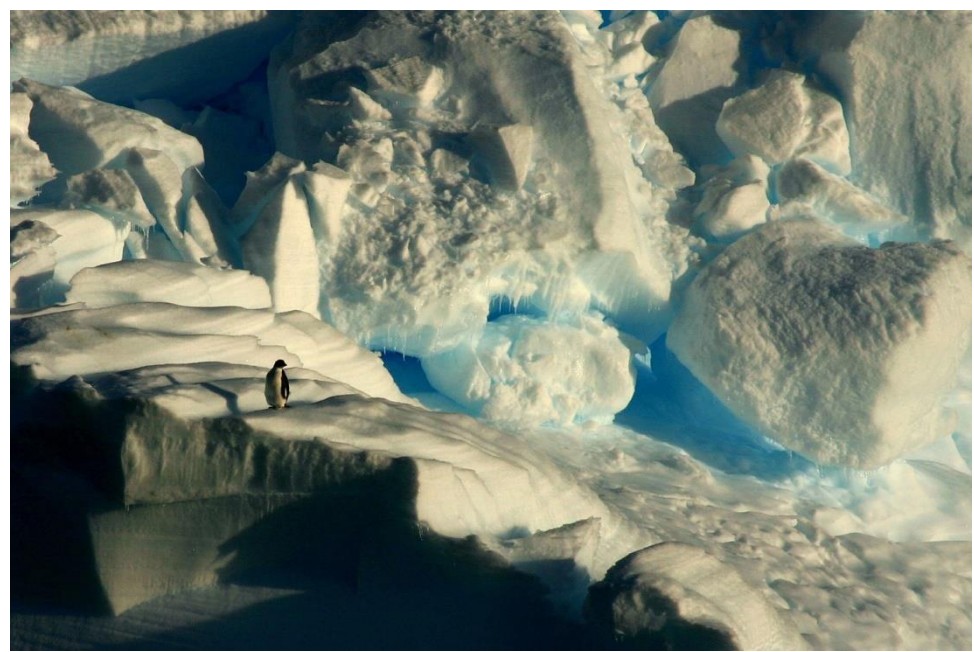

Figure 7: A lonely penguin on a drifting iceberg (Ross Sea). XXI PNRA Antarctic expedition, project WISE.

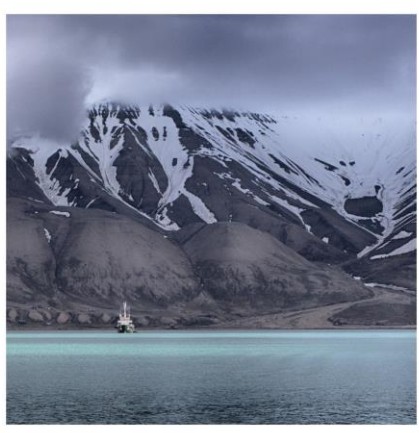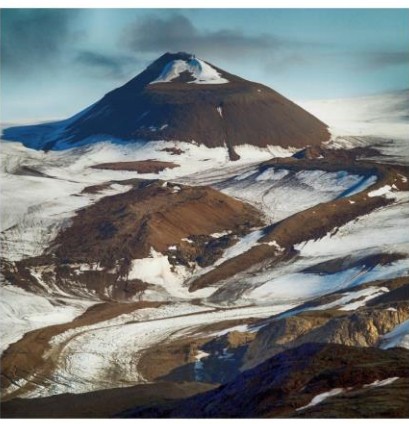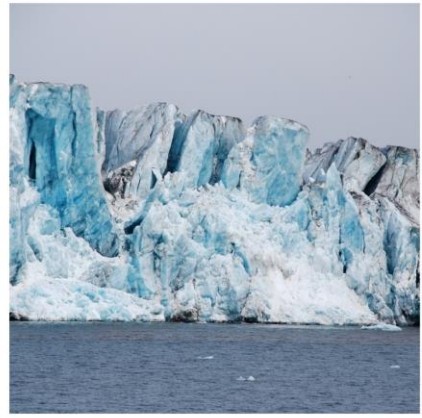

Figure 8: Svalbard landscapes (Svalbard archipelago, Norway). a) Longyearbyen Bay, Tundra landform. R/V Polarstern expedition PS99-1a, Eurofleets2 project "*Bottom currents in a stagnant environment- BURSTER*"; b) A view from the Wordiekammen plateau towards the Ebbabreen, with the nunatak Bastonfjellet, Paleokarst project; c) Front of the Bellsund ice stream (SW Svalbard). RV Ian Mayen 2009 expedition, University of Tromsø-UiT, "*Glaciations in the Barents Sea Area –GLACIBAR*" project; d) From the Wordiekammen plateau towards the Petunia Bukta, with the waters from Horbyedalen and Ebbadalen, Paleokarst project.

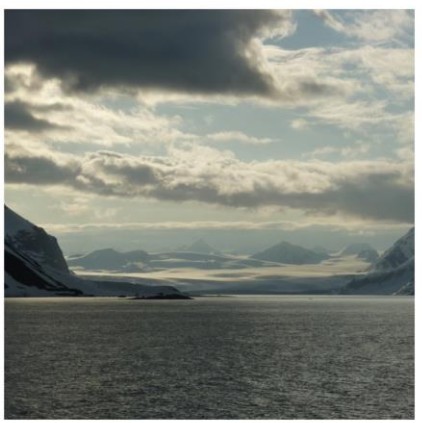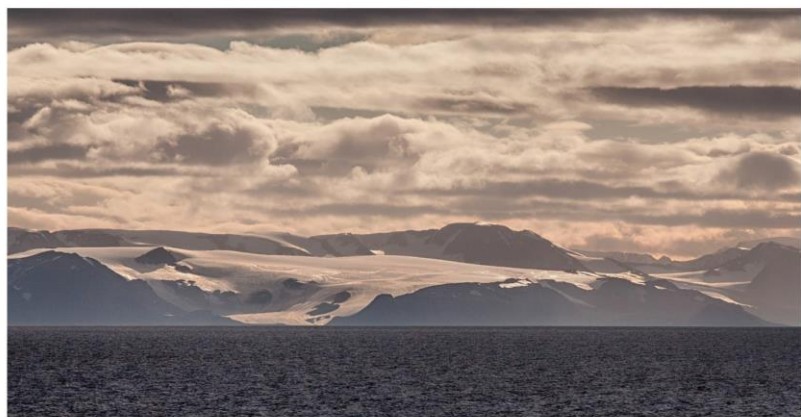

Figure 9: Svalbard landscapes. a) Hornsund Fjord, Spitsbergen. RV G.O. Sars, expedition 191, "*Present and past flow regime on contourite drifts west of* Spitsbergen *Area- PREPARED*" Eurofleets2 project; b) Ice coverage of the Svalbard 'Islands' northwestern coast. R/V OGS Explora, "*Petroleum Assessment of the Arctic North Atlantic and adjacent marine areas- PANORAMA*" project; c) Skanskbukta Bay (on the left), Billefjorden (centre) with Bünsow Land cliffs (front). Field trip "*Poli Arctici Skanskbukta basecamp*", the "*Northern Rangers*" group.

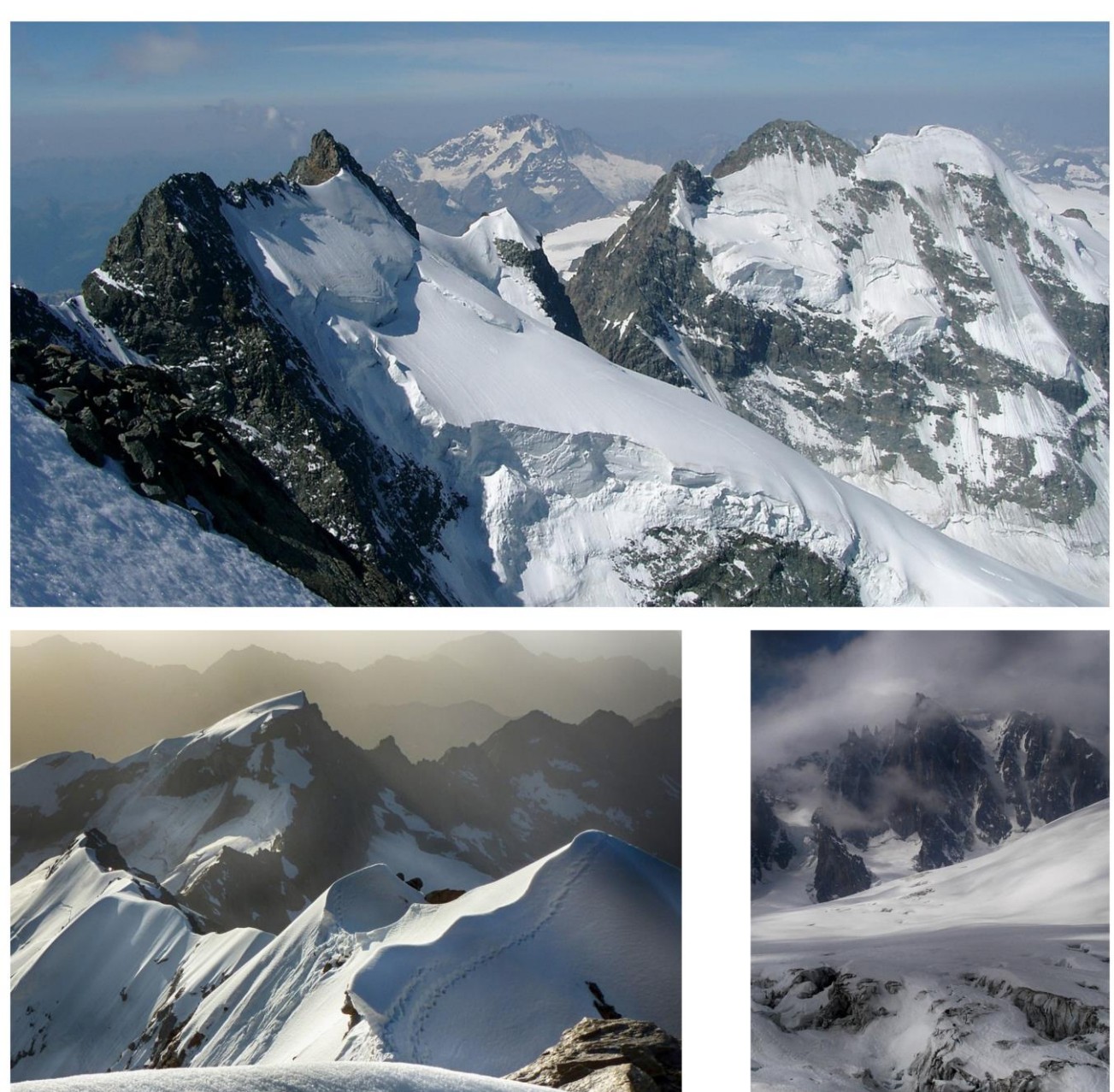

Figure 10: Mountain landscapes. a) Piz Bernina (Italy); b, c) PNRA-WISSLAKE project: b) Monte Rosa (Italy); c) Giant Glacier, Mont Blanc (Italy).

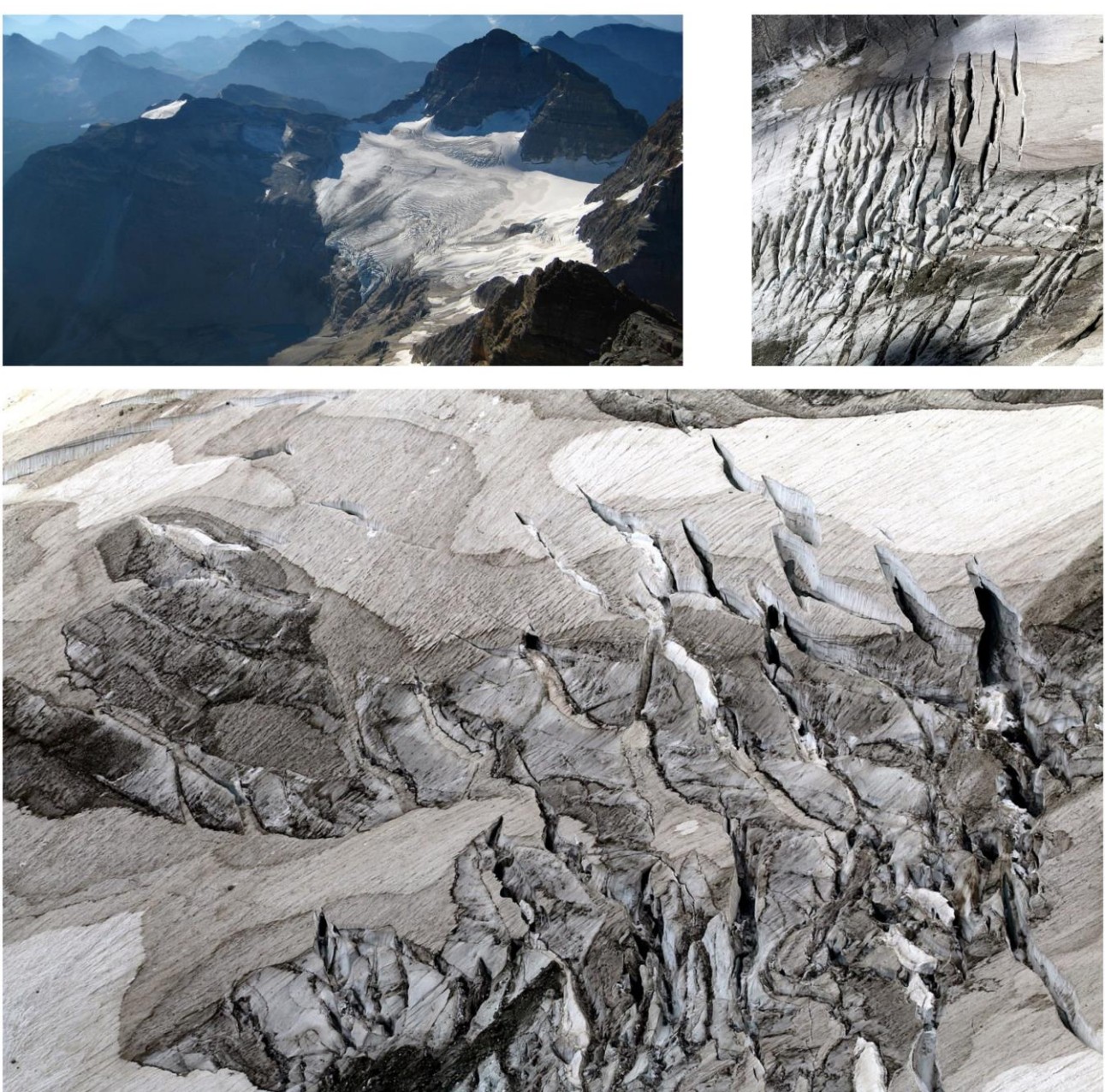

Figure 11: a) A glacier on Mount Assiniboine, British Columbia, Canada. Field trip in the framework of the SEG 2009 Summer Research Workshop on "*CO₂ Sequestration Geophysics*"; b, c) A minor glacier in the Mont Blanc group (Italy). Field trip in the frame of the Near Surface Geoscience 2015 - 21$^{st}$ European Meeting of Environmental and Engineering Geophysics.