# Peer review of ""Focus on glaciers": a geo-photo exposition of vanishing beauty."

_Geoscience Communication, 2020_

## Referee Comment (RC1) · Anonymous Referee #1 · 29 Apr 2020

General Comments The manuscript is focused on an actual topic: fostering public engagement and awareness of climate change risks through a photographic exhibition of glaciers. Particular attention has been given to the communication strategy, as well as to pictures selection. The choice to prefer positive and hope messages, although considering the seriousness of the topic, is an effective strategy. In general, the research approach is quite good, the topic is well introduced, and the methodology of the work is clearly shown. The research methodology and the results are convincing. Following comments might help improve the quality of the manuscript.

Specific comments 1. Does the paper address relevant scientific questions within the scope of GC? Yes, this contribution fits with GC aims and scope, since the authors' whish have been the communication of a particular aspect of climate change, through

a strict connection between science and art. 2. Are the scientific methods and assumptions valid and clearly outlined? Yes, the strategy of communication, engagement and selection of the pictures for the exhibition is clearly reported, as well as the audience response expected from the authors. Also, the relation between the pictures, the places where they have been taken and the specific issues of each are well reported. A suggestion would be citing in paragraph 4 the initial number of pictures submitted for the internal call (before the selection of the final 26 for the exhibition), in order to understand the real participation to the initiative. 3. Are the results sufficient to support the interpretations and conclusions? Yes, the good number of visitors and adopted communication strategies seem to confirm the engagement and vehiculation of the message the authors wanted to transmit through the pictures. 4. Do the authors give proper credit to related work and clearly indicate their own new/original contribution? Does the paper present novel concepts, ideas, tools, or data? It can be improved. The original contribution is not very clear: the organization of an exhibition of pictures, independently on the topic, is not an innovative approach for communication. The authors should stress more the attention given to details, such as public engagement and feelings. 5. Does the title clearly reflect the contents of the paper? Yes. 6. Does the abstract provide a concise and complete summary? Yes but the language style is too informal 7. Is the overall presentation well structured and clear? Yes, the aim of the authors is clear and understandable. 8. Is the language fluent and precise? It has to be improved. The overall style of the entire manuscript is too informal and double check typos and grammar errors over the entire manuscript is strongly suggested. 9. Are the number and quality of references appropriate? Yes. The topics related to climate change are usually well referenced in the manuscript. Also, science and communication have some references. I suggest adding a reference to European Agenda 2030 and the Goals for Sustainable Developments in paragraph 1.

---

## Referee Comment (RC2) · Mariele Neudecker (Referee) · 10 May 2020

Overall, I really appreciate the writing of a thesis about the subject. This is written from the perspective of a visual artist, not a scientist.

I would imagine that this text would include some more detail and comparison of the subject and also of the urgently needed steps for us all to take. It is one side of the issue to document "so-called: vanishing beauty", it is really key to also to show the other side and document the wrong-doings of humanity, for example the way food gets produced, handled, shipped and distributed; also the way be deal with waste of our products etc. transport and travel, etc.

I have learned to be very sceptical of the word "beauty" and it does need definition.

[Figure]

Equally I have learned to use the word "wilderness" very carefully and again, clarify what it meant, especially with this discussion.

I would not use the word "picture" and replace it with "document, photograph or image".

I would look at for example at the organisation Project Pressure and make various international comparisons as there are various groups doing this kind of documentation. Compare notes?! E.G.: https://www.project-pressure.org/ (quote from website: Since 2008 Project Pressure has been commissioning world-renowned artists to conduct expeditions around the world for the purpose of creating an exhibition visualizing the climate crisis.) It would be good to show awareness of this and see the bigger picture in the thesis and investigate that in a critical way.

The loss of mass with glaciers would be important to visualise, hence to show some in comparison would be essential? Somehow the last sentence is asking for more: we need to see the climate crisis, we need to understand the problems and we all need to be aware what to do, what is it to do first? I am saying this provocatively I hope, as a lot of us do know, and still don't do it.

Is anyone with English as first language proof-reading this? – as I am finding quite a few spelling or choice of word mistakes. (otherwise I could do some with this, but English is not my 1st language either).

(nothing attached for now)

---

## Author Comment (AC1) · 17 Jun 2020

We kindly acknowledge the reviewer for his/her time, accurate reading, appreciation, and valuable comments that will be of help in improving our manuscript. In the following, we reply to his/her comments point by point (you find the same in the attached file).

1) Does the paper address relevant scientific questions within the scope of GC? Yes, this contribution fits with GC aims and scope, since the authors' whish have been the communication of a particular aspect of climate change, through a strict connection between science and art.

Thank you for your appreciation

2) Are the scientific methods and assumptions valid and clearly outlined? Yes, the strategy of communication, engagement, and selection of the pictures for the exhibition is clearly reported, as well as the audience response expected from the authors. Also, the relation between the pictures, the places where they have been taken, and the specific issues of each are well reported. A suggestion would be to cite in paragraph 4 the initial number of pictures submitted for the internal call (before the selection of the final 26 for the exhibition), in order to understand the real participation to the initiative.

Thank you for the good suggestion. We shall add the number in the text. We received 130 photographs, among which we chose about 20% for the exhibition.

3) Are the results sufficient to support the interpretations and conclusions? Yes, the good number of visitors and adopted communication strategies seem to confirm the engagement and vehiculation of the message the authors wanted to transmit through the pictures.

Thank you for this appreciation.

4) Do the authors give proper credit to related work and clearly indicate their own new/original contribution? Does the paper present novel concepts, ideas, tools, or data? It can be improved. The original contribution is not very clear: the organization of an exhibition of pictures, independently on the topic, is not an innovative approach for communication. The authors should stress more the attention given to details, such as public engagement and feelings.

We will clarify better in the manuscript what the originality of our contribution consists of. We agree with the reviewer, it is certainly not a novelty to organize exhibitions to communicate. Several photo exhibitions have been organized during these years by professional photographers and artists within projects devoted to enlarge the public awareness on this theme, using the art to strengthen the message( e.g., https://sulletraccedeighiacciai.com, https://www.project-pressure.org/mariele-neudecker-and-project-pressure-partnership/). We shall add

Interactive
comment

some reference to these initiatives in the revised manuscript, underlining their importance. As a specific example of initiatives aimed at integrating art and science, we will also include the Extreme Ice Survey program (http://extremeicesurvey.org/), which produced a photography book (Balog et al., 2012) and a documentary film, "Chasing Ice" (https://chasingice.com/), winner of an Emmy Award in 2014. However, it is unusual for scientists to organize exhibitions, as we did, making available the materials collected during scientific campaigns for study purposes different than the themes of the exhibition, thanks to the personal sensibility of the authors of the pictures. As reported in the article, all the authors are scientists involved in scientific activities on research cruises and not professional photographers. Some other online collections from scientists are available, like the one managed by the National Snow and Ice Data Center, https://nsidc.org/data/glacier_photo/, or the "Repeat Photography Project" of the USGS Northern Rocky Mountain Science Center, focussed on the Glacier National Park, https://www.usgs.gov/centers/norock/science/repeat-photography-project?qt-science_center_objects=0#qt-science_center_objects. Another repository of pictures on various geoscience themes shot by the scientists is the images archive of EGU (https://imaggeo.egu.eu/). In this case, the archive is accessed by the scientific community, although geosciences involve a vast community. Only in some cases, the best photos, awarded during the annual conference, are printed as cards and reach a wider public. Our goal, on the contrary, was to fill the gap between research and society: the exhibition becomes the way to bring scientists near the public, and specifically, adult people, in working age, in an environment extraneous to science. The venue, in fact, was chosen among the places not usually used for scientific dissemination activities as the ones used for Science Cafè or conferences, but it was the hall of a chamber of commerce usually crowded during working hours. We wanted to talk about science, describing where the photos were taken, in which conditions, for which specific research project. In fact, some of us received many technical questions not only on climate change but on the geology and geomorphology of glaciers as well, thus adding value and a scientific significance to the artistic quality of
the images. This experience may be further stimulated within the research community, also to keep track and record of the fast changes occurring in the global glaciers, as well as finding among our pictures other themes to be exposed in similar exhibitions. We propose to add such considerations to clarify the originality of our contribution. We shall also add that the exhibition is now permanent in OGS premises, visible to all our visitors and collaborators. Balog, J. (2012) Ice: Portraits of Vanishing Glaciers, Rizzoli, 288 pp.

8) Is the language fluent and precise? It has to be improved. The overall style of the entire manuscript is too informal and double-check of typos and grammar errors over the entire manuscript is strongly suggested.

To ensure an adequate level of English, we intend to work with a professional English editing service to improve the overall style of the final version of the manuscript.

9) Are the number and quality of references appropriate? Yes. The topics related to climate change are usually well referenced in the manuscript. Also, science and communication have some references. I suggest adding a reference to the European Agenda 2030 and the Goals for Sustainable Developments in paragraph 1.

Thank you for this note. We will add in the manuscript the following reference : United Nations (2015). Transforming our World: The 2030 Agenda for Sustainable Development. A/RES/70/1 41 pp. https://sustainabledevelopment.un.org/post2015/transformingourworld/publication
We will also refer to the SDG n. 13 "Climate Action", specific target 13.3 "Improve education, awareness-raising and human and institutional capacity on climate change mitigation, adaptation, impact reduction and early warning", addressed by our exhibition.

Please also note the supplement to this comment:
https://gc.copernicus.org/preprints/gc-2020-3/gc-2020-3-AC1-supplement.pdf

---

## Author Comment (AC2) · 17 Jun 2020

Dear Dr. Neudecker, we kindly acknowledge you for your time, the critical reading of our work, and the valuable comments, criticisms, and suggestions that will be of help in improving our manuscript. In the following, we reply to your comments, point by point (you find the same in the attached file).

1) I would imagine that this text would include some more detail and comparison of the subject and also of the urgently needed steps for us all to take. It is one side of the issue to document "so-called: vanishing beauty", it is really key to also to show the other side and document the wrong-doings of humanity, for example, the way food gets produced, handled, shipped and distributed; also the way we deal with waste of our

products, etc. transport and travel, etc.

Our concepts of world and environment must necessarily change for a new coexistence of human society and nature. OGS is significantly involved in educational and outreach activities aimed at increasing awareness in public about the environmental impacts in the ocean (e.g., pollution, plastic, overfishing), and we choose to use different communication strategies to convey this message. The purpose of this particular exhibition was to attract the interest of the general public to environmental problems and, in this case, not to teach and disseminate good practices for our everyday life, what we do on other occasions.

2) I would look at for example at the organisation Project Pressure and make various international comparisons as there are various groups doing this kind of documentation. Compare notes?! E.G.: https://www.project-pressure.org/ (quote from website: Since 2008 Project Pressure has been commissioning world-renowned artists to conduct expeditions around the world for the purpose of creating an exhibition visualizing the climate crisis.) It would be good to show awareness of this and see the bigger picture in the thesis and investigate that in a critical way.

Thank you for your valuable suggestion and your involvement in such an interesting project. Art and science events, as you point out, are not a novelty and we are aware that several photo exhibitions have been organized during the last years by professional photographers and artists within projects devoted to enlarge the public awareness on this theme, using the art to strengthen the message ( e.g., https://sulletraccedeighiacciai.com , http://www.project-pressure.org). As additional examples of initiatives aimed at integrating art and science, we will also include the Extreme Ice Survey program (http://extremeicesurvey.org/), which produced a photography book (Balog et al., 2012) and a documentary film, "Chasing Ice" (https://chasingice.com/), winner of an Emmy Award in 2014. We will add citations to all these activities in the manuscript, and we will better clarify that it is unusual for scientists to organize exhibitions, as we did, making

available the materials collected during scientific campaigns for study purposes different than the themes of the exhibition, thanks to the personal sensibility of the authors of the pictures. Some other online collections of pictures from scientists are available, but the archives are known (and accessed) by the scientific community only and not public at large (e.g.,https://nsidc.org/data/glacier_photo/, https://www.usgs.gov/centers/norock/science/repeat-photography-project?qt-science_center_objects=0#qt-science_center_objects, https://imaggeo.egu.eu/). A different example is given by the way some scientists succeed in visualizing their data, so to make them almost artistic: it is the case of Ed Hawkins (National Centre for Atmospheric Science at the University of Reading - https://earther.gizmodo.com/this-climate-visualization-belongs-in-a-damn-museum-1826307536), Antti Lipponen (Finnish Meteorological Institute- https://earther.gizmodo.com/a-new-visualization-turns-global-warming-into-pop-art-1828625479 ). Our goal was to fill the gap between research and society: the exhibition becomes the way to bring scientists closer to the public, and precisely, adult people, in working age, in an environment extraneous to science. The venue, in fact, was chosen among the places not usually used for scientific dissemination activities as the ones used for Science Cafè or conferences, but it was the hall of a chamber of commerce usually crowded during working hours. We wanted also to talk about science, describing where the photos were taken, in which conditions, for which specific research project. In fact, some of us received many technical questions not only on climate change but on the geology and geomorphology of glaciers as well, thus adding value and a scientific significance to the artistic quality of the images. This experience may be further stimulated within the research community, also to keep track and record of the fast changes occurring in the global glaciers, as well as finding among our pictures other themes to be exposed in similar exhibitions.

3) The loss of mass with glaciers would be important to visualise, hence to show some in comparison would be essential?

[Figure]

We agree with you that having the photo comparison of the same glaciers over the years is of excellent communication impact (as already done in other initiatives, as https://sulletraccedeighiacciai.com or "Chasing Ice" - https://chasingice.com/)). As our exhibition is an a-posteriori collection of photos shot during short term scientific OGS campaigns for study purposes different than the time-lapse documentation of melting glaciers, it was impossible for us to document the transformation over the years of the different places. However, we believe worth exploiting the significant number of our pictures to witness the grandeur of a landscape that is in danger of extinction. We will better clarify this point in the manuscript.

4) Somehow the last sentence is asking for more: we need to see the climate crisis, we need to understand the problems and we all need to be aware what to do, what is it to do first? I am saying this provocatively I hope, as a lot of us do know, and still don't do it.

Disappearing of glaciers is a piece of striking evidence that global warming is happening here and now, and will (probably) profoundly affect how our entire society will function in the future. Global warming is an entity of such vast temporal and spatial dimensions, so interconnected with all human activities, that seems to defy not only our control but also our understanding. Our concepts of world and environment must necessarily change for a new coexistence of human society and nature. Communication activities like our exhibition and other actions we, at OGS, and others are doing, are vital to highlight the problem and make it relevant to the general public. The debate about climate change communication strategies is still active, and catastrophic frames are controversial (see Public Understanding of Science 2019, Vol. 28(4) 401–416). The exhibition project is still ongoing: pictures are now exposed at OGS premises, and our colleagues are encouraged to collect new material during scientific expeditions to propose updated versions of the exhibition. In future events, we shall try to further involve the visitors, through short surveys, aimed to verify whether the message has passed, and the awareness level has increased after visiting the exhibition.

**[GCD](GCD)**
Please also note the supplement to this comment:
https://gc.copernicus.org/preprints/gc-2020-3/gc-2020-3-AC2-supplement.pdf

---

## Author Response (AR1)

**Giuliana Rossi (e-mail: grossi@inogs.it)**
**National Institute of Oceanography**
**and Applied Geophysics -OGS**
**Sezione Center for Seismological**
**Research**
Borgo Grotta Gigante 42/c- 34010 Sgonico
(Trieste) Italy

Trieste, September 3rd 2020

**Editorial Office**
**Geoscience Communications**
**To the attention of Dr Francesco**
**Mugnai**

**Object: Manuscript submission**

Dear Editor,

We submit the revised version of the paper GC2020-3 "Focus on glaciers": a geo-photo exposition of vanishing beauty" for the special issue issue "Five years of Earth sciences and art at the EGU (2015– 2019)".

We thank you, Mariele Neudecker and the other reviewer for the constructive and stimulating comments, which led to significant improvements of this paper.

We rewrote almost completely the paper, and we let it be reviewed by professionals in language editing, to address the comments of both reviewers, and in particular, points 6 and 8 of reviewer 1. As regards as reviewer 1, we added the reference to Agenda 2030, and stressed the peculiarity of our exhibition, if compared with other visual initiatives, by adding details on the photograph choice, the public engagement, the feeling of both exhibitors and visitors.

As regards as the stimulating comments of reviewer 2, Mariele Neudecker, we added more references to other important initiatives as Project Pressure, Chasing Ice, and others, to show the context in which our exhibition lays and the differences. In particular, we tried to explain the importance and value of our exhibition in involving people in the delicate theme of the climate crisis, through high-impact photographs, although for us it was not possible to have time-lapse images of the same subject, being almost all bound to single campaigns, rarely, if never, in the same places.

We also corrected the captions, adding more details on the projects, and corrected Figure 3, since there was a mistyping mistake. We uploaded the revised figures as supplementary material.

Kind regards

Giuliana Rossi, on behalf of the co-authors

**Reviewer 1:**
We kindly acknowledge the reviewer for his/her time, accurate reading, appreciation, and valuable comments that have been of help in improving our manuscript. In the following, we reply to his/her comments point by point.

1) ***Does the paper address relevant scientific questions within the scope of G.C.?***

*Yes, this contribution fits with G.C. aims and scope, since the authors' whish have been the communication of a particular aspect of climate change, through a strict connection between science and art.*

2) ***Are the scientific methods and assumptions valid and clearly outlined?***
*Yes, the strategy of communication, engagement, and selection of the pictures for the exhibition is clearly reported, as well as the audience response expected from the authors. Also, the relation between the pictures, the places where they have been taken, and the specific issues of each are well reported.*
*A suggestion would be citing in paragraph 4 the initial number of pictures submitted for the internal call (before the selection of the final 26 for the exhibition), in order to understand the real participation to the initiative.*

Thank you for the excellent suggestion. We added the number in the text. We received 130 photographs, among which we chose about 20% for the exhibition.

3) ***Are the results sufficient to support the interpretations and conclusions?***
*Yes, the good number of visitors and adopted communication strategies seem to confirm the engagement and vehiculation of the message the authors wanted to transmit through the pictures.*

Thank you for this appreciation.

4) ***Do the authors give proper credit to related work and clearly indicate their own new/original contribution? Does the paper present novel concepts, ideas, tools, or data?***
*It can be improved. The original contribution is not very clear: the organisation of an exhibition of pictures, independently on the topic, is not an innovative approach for communication. The authors should stress more the attention given to details, such as public engagement and feelings.*

We clarified better in the manuscript what the originality of our contribution consists of. We agree with the reviewer that it is certainly not a novelty to organise exhibitions to communicate. Several photo exhibitions have been organised during these years by professional photographers and artists within projects devoted to enlarge the public awareness on this theme, using the art to strengthen the message (e.g., https://sulletraccedeighiacciai.com, https://www.project-pressure.org/mariele-neudecker-and-project-pressure-partnership/). We shall add some reference to these initiatives in the revised manuscript, underlining their importance. As a specific example of initiatives aimed at integrating art and science, we also included the Extreme Ice Survey program (http://extremeicesurvey.org/), which produced a photography book (Balog et al., 2012) and a documentary film, "Chasing Ice" (https://chasingice.com/), winner of an Emmy Award in 2014.
However, it is unusual for scientists to organise exhibitions, as we did, making available the materials collected during scientific campaigns for study purposes different than the themes of the exhibition, thanks to the personal sensibility of the authors of the pictures.  As

reported in the article, all the authors are scientists involved in scientific activities on research cruises and not professional photographers. Some other online collections from scientists are available, like the one managed by the National Snow and Ice Data Center, https://nsidc.org/data/glacier_photo/, or the "Repeat Photography Project" of the USGS Northern Rocky Mountain Science Center, focussed on the Glacier National Park, https://www.usgs.gov/centers/norock/science/repeat-photography-project?qt-science_center_objects=0#qt-science_center_objects. Another repository of pictures on various geoscience themes shot by the scientists is the images archive of EGU (https://imaggeo.egu.eu/). In this case, the archive is accessed by the scientific community, although geosciences involve a vast community. Only in some cases, the best photos, awarded during the annual conference, are printed as cards and reach a wider public. Our goal, on the contrary, was to fill the gap between research and society: the exhibition becomes the way to bring scientists near the public, and precisely, adult people, in working age, in an environment extraneous to science. The venue was chosen among the places not usually used for scientific dissemination activities as the ones used for Science Cafè or conferences, but it was the hall of a chamber of commerce usually crowded during working hours. We wanted to talk about science, describing where the photos were taken, in which conditions, for which specific research project. Some of us received many technical questions not only on climate change but on the geology and geomorphology of glaciers as well, thus adding value and a scientific significance to the artistic quality of the images. This experience may be further stimulated within the research community, also to keep track and record of the fast changes occurring in the global glaciers, as well as finding among our pictures other themes to be exposed in similar exhibitions.

We added such considerations to clarify the originality of our contribution. We also added that the exhibition is now permanent in OGS premises, visible to all our visitors and collaborators.

Balog, J. (2012) Ice: Portraits of Vanishing Glaciers, Rizzoli, 288 pp.

*8) Is the language fluent and precise?*
*It has to be improved. The overall style of the entire manuscript is too informal and double-check typos and grammar errors over the entire manuscript is strongly suggested.*

To ensure an adequate level of English, we used a professional English editing service to improve the overall style of the final version of the manuscript.

*9) Are the number and quality of references appropriate?*
*Yes. The topics related to climate change are usually well referenced in the manuscript. Also, science and communication have some references. I suggest adding a reference to the European Agenda 2030 and the Goals for Sustainable Developments in paragraph 1.*

Thank you for this note. We added in the manuscript the following reference :
United Nations: Transforming our World: The 2030 Agenda for Sustainable Development. A/RES/70/1  41 pp., 2015.
https://sustainabledevelopment.un.org/post2015/transformingourworld/publication
We also referred to the SDG n. 13 "Climate Action", specific target 13.3 "Improve education, awareness-raising and human and institutional capacity on climate change mitigation, adaptation, impact reduction and early warning", addressed by our exhibition.

**Reviewer 2 Mariele Neudecker**

Dear Dr. Neudecker,
We kindly acknowledge you for your time, the critical reading of our work, and the valuable comments, criticisms, and suggestions that have been of help in improving our manuscript. In the following, we reply to your comments, point by point.

1)      *I would imagine that this text would include some more detail and comparison of the subject and also of the urgently needed steps for us all to take. It is one side of the issue to document "so-called: vanishing beauty", it is really key to also to show the other side and document the wrong-doings of humanity, for example, the way food gets produced, handled, shipped and distributed; also the way we deal with waste of our products, etc. transport and travel, etc.*

Our concepts of world and environment must necessarily change for a new coexistence of human society and nature. OGS is significantly involved in educational and outreach activities aimed at increasing public awareness about the environmental impacts in the ocean (e.g., pollution, plastic, overfishing) and we choose to use different communication strategies to convey this message. The purpose of this particular exhibition was to attract the interest of the general public to environmental problems and, in this case, not to teach and disseminate good practices for our everyday life, what we do on other occasions.

2)      *I would look at, for example, the organisation Project Pressure and make various international comparisons as there are various groups doing this kind of documentation. Compare notes?! E.G.: https://www.project-pressure.org/ (quote from the website: Since 2008, Project Pressure has been commissioning world-renowned artists to conduct expeditions around the world for the purpose of creating an exhibition visualising the climate crisis.) It would be good to show awareness of this and see the bigger picture in the thesis and investigate that in a critical way.*

Thank you for your valuable suggestion and your personal involvement in such an exciting project. Art and science events, as you point out, are not a novelty and we are aware that several photo exhibitions have been organised during the last years by professional photographers and artists within projects devoted to enlarge the public awareness on this theme, using the art to strengthen the message (e.g.,  https://sulletraccedeighiacciai.com , http://www.project-pressure.org). As additional examples of initiatives aimed at integrating art and science, we will also include the Extreme Ice Survey program (http://extremeicesurvey.org/), which produced a photography book (Balog et al., 2012) and a documentary film, "Chasing Ice" (https://chasingice.com/), winner of an Emmy Award in 2014. We added citations to all these activities in the manuscript, and we better clarified that it is unusual for scientists to organise exhibitions, as we did, making available the materials collected during scientific campaigns for study purposes different than the themes of the exhibition, thanks to the personal sensibility of the authors of the pictures.  Some other online collections of pictures from scientists are available, but the archives are known (and accessed) mostly by the scientific community only and not public at large (e.g., https://nsidc.org/data/glacier_photo/, https://www.usgs.gov/centers/norock/science/repeat-photography-project?qt-science_center_objects=0#qt-science_center_objects, https://imaggeo.egu.eu/).
A different example is given by the way some scientists succeed in visualising their data, so to make them almost artistic: it is the case of Ed Hawkins (National Centre for Atmospheric Science at the University of Reading - https://earther.gizmodo.com/this-climate-visualization-belongs-in-a-damn-museum-1826307536), Antti Lipponen (Finnish

Meteorological Institute- https://earther.gizmodo.com/a-new-visualization-turns-global-warming-into-pop-art-1828625479 ). Our goal was to fill the gap between research and society: the exhibition becomes the way to bring scientists closer to the public, and precisely, adult people, in working age, in an environment extraneous to science. The venue was chosen among the places not usually used for scientific dissemination activities as the ones used for Science Cafè or conferences, but it was the hall of a chamber of commerce usually crowded during working hours. We also wanted to talk about science, describing where the photos were taken, in which conditions, for which specific research project. Some of us received many technical questions not only on climate change but on the geology and geomorphology of glaciers as well, thus adding value and a scientific significance to the artistic quality of the images. This experience may be further stimulated within the research community, also to keep track and record of the fast changes occurring in the global glaciers, as well as finding among our pictures other themes to be exposed in similar exhibitions.

3)      *The loss of mass with glaciers would be important to visualise, hence to show some in comparison would be essential?*

We agree with you that having the photo comparison of the same glaciers over the years is of high communication impact (as already done in other initiatives, as https://sulletraccedeighiacciai.com or  "Chasing Ice" - https://chasingice.com/)).
As our exhibition is an a-posteriori collection of photos shot during short term scientific OGS campaigns for study purposes different than the time-lapse documentation of melting glaciers, we could not document the transformation over the years of the different places. However, we believe worth exploiting the large number of OGS's photographs to witness the grandeur of a landscape that is in danger of extinction.  We will better clarify this point in the manuscript.

4)      *Somehow the last sentence is asking for more: we need to see the climate crisis, we need to understand the problems and we all need to be aware what to do, what is it to do first? I am saying this provocatively I hope, as a lot of us do know, and still don't do it.*

Disappearing of glaciers is a piece of striking evidence that global warming is happening here and now, and will (probably) profoundly affect how our entire society will function in the future. Global warming is an entity of such vast temporal and spatial dimensions, so interconnected with all human activities, that seems to defy not only our control but also our understanding. Our concepts of world and environment must necessarily change for a new coexistence of human society and nature. Communication activities as our exhibition and other actions we, at OGS, and others are doing, are essential to highlight the problem and make it relevant to the general public. The debate about climate change communication strategies is still active, and catastrophic frames are controversial (see Public Understanding of Science 2019, Vol. 28(4) 401–416).
The exhibition project is still ongoing: pictures are now exposed at OGS premises, and our colleagues are encouraged to collect new material during scientific expeditions to propose updated versions of the exhibition. In future events, we shall try to further involve the visitors, through short surveys, aimed to verify whether the message has passed, and the awareness level has increased after visiting the exhibition.
We added these considerations in the paper.

[revised manuscript text omitted]
); or the lonely penguin of set on a drifting iceberg (Figure 7, a) as an emblematic symbol of all the animalliving species in danger of extinction due to the climate crisis. The picture of Figure 8d and the graphicgraphical effects shown intransmitted by Figures 11 b, c well represent the11b and 11c dramatically document glacier melting and the possible desolation of the future aspectlandscape. The multiple visioncontemplation of the 26 picturesphotographs as a whole produced a strengthening of the *message* that the viewer perceives,perceived even in a fleeting passage inthrough a public, crowded place.

The Therefore, the exhibition was opened in 2016, from October 17th to October 31st (i.e., one week beyond the end ofbecame the "*Settimana del pianeta Terra" - The Week of Earth planet*).way to bring scientists closer to the public, taking specifically into consideration the working-age adults (18-64 years) in an environment typically unrelated to science. The location seat of the lobby of the Chamber of Commerce of Trieste wasappeared to be an excellent choice: about approximately 100

[revised manuscript text omitted]

environment closestcourse of future events, we will further involve visitors through short surveys to us,verify whether the Alps, helps to make transmitted message was easily accessible and the researcher experiences nearer tolevel of awareness obtained after the onesvisit 
[revised manuscript text omitted]